# *In Vitro* Interaction of Binuclear Copper Complexes with Liver Drug-Metabolizing Cytochromes P450

**DOI:** 10.3390/ph17091194

**Published:** 2024-09-10

**Authors:** Alena Špičáková, Zuzana Horáčková, Pavel Kopel, Pavel Anzenbacher

**Affiliations:** 1Department of Pharmacology, Faculty of Medicine, Palacký University Olomouc, Hněvotínská 3, 779 00 Olomouc, Czech Republic; anzen@seznam.cz; 2Laboratory of Growth Regulators, Faculty of Science, Palacký University & Institute of Experimental Botany of the Czech Academy of Sciences, Šlechtitelů 27, 78 371 Olomouc, Czech Republic; zuzi.horackova@seznam.cz; 3Department of Inorganic Chemistry, Faculty of Science, Palacký University Olomouc, 17. Listopadu 1192/12, 779 00 Olomouc, Czech Republic

**Keywords:** cytochromes P450, enzyme activity, inhibition, drug interactions, copper complexes, dicarboxylic acid, antibacterial activity

## Abstract

Two copper(II) mixed ligand complexes with dicarboxylate bridges were prepared and studied, namely [Cu_2_(μ-fu)(pmdien)_2_(H_2_O)_2_](ClO_4_)_2_ (complex No. 5) and [Cu_2_(μ-dtdp)(pmdien)_2_(H_2_O)_2_](ClO_4_)_2_ (complex No. 6), where H_2_fu = fumaric acid, pmdien = *N,N,N′,N″,N″* pentamethyldiethylenetriamine, and H_2_dtdp = 3,3′-dithiodipropionic acid. The copper atoms are coordinated in the same mode by the tridentate pmdien ligand and oxygen of water molecules, and they only differ in the dicarboxylate bridge. This work is focused on the study of the inhibitory effect of these potential antimicrobial drugs on the activity of the most important human liver drug-metabolizing enzymes, cytochromes P450 (CYP), especially their forms CYP2C8, CYP2C19, and CYP3A4. The obtained results allow us to estimate the probability of potential drug interactions with simultaneously administrated drugs that are metabolized by these CYP enzymes. In conclusion, the presence of adverse effects due to drug–drug interactions with concomitantly used drugs cannot be excluded, and hence, topical application may be recommended as a relatively safe approach.

## 1. Introduction

Copper is an essential element for the human organism [1]. It fulfills important roles in many biological processes as an integral part of proteins, helping in maintaining the structural properties determining their function [2]. As an example, its role in the terminal oxidase of oxidative phosphorylation, in binding and in activation of dioxygen to prevent the formation of relative oxygen species can be mentioned [1]. Deficient transport of copper is known to cause diseases such as Wilson´s disease (resulting in the accumulation of copper in the liver and other organs causing, e.g., corneal deposition and neurological symptoms) or Menkes´ disease (with accumulation of copper in the intestine and its deficiency in the rest of the organism leading to neurological degeneration and other pathological disorders [3]). The ability of copper to bind to various structures and form complexes acting as chemotherapeutics has recently been reviewed [4].

Organometalic compounds with copper and various ligands have interested many researchers for years. The carboxylate anions can be bonded to central atoms by different coordination modes, affecting the formed species’ nuclearity. The complexes that are formed in this manner can be interesting from many points of view. There are optical [5], magnetic [6,7], and catalytic properties, which are investigated [8,9]. Moreover, carboxylic acids are often utilized for the preparation of metal–organic frameworks (MOFs). The bridges can be formed by carboxylates, but very often, they are formed by a combination of carboxylate and suitable nitrogen donor ligands, which can extend the distances among central metal ions. MOFs are usually studied for their intriguing physical properties and formation of cavities of different shapes and sizes, but they have also been applied as antibacterial agents or drug carriers [10,11].

The biological properties of coordination compounds with carboxylates, as systems mimicking enzymes and processes in living organisms, are often studied. It is necessary to mention the potential of such compounds to be exploited as drugs to treat many diseases, including cancer, and, of course, their potential antibacterial activity, especially against drug-resistant bacteria species. For example, a binuclear Cu(II) complex with a 2,6-pyridinedicarboxylate bridge and 2-aminopyridine was evaluated for antimicrobial and radical scavenging activities [12]. Cobalt(III) complexes with pyrazine-2,3-dicarboxylate and tetramethylethylenediamine and 2,2-dimethylpropane-1,3-diamine were prepared and characterized, and their antimicrobial activity was reported by Yesiel et al. [13]. Silver dicarboxylate complexes with 1,10-phenanthroline were synthesized, and their high solubility and chemotherapeutic potential against several bacteria and cancer cells were reported by Thornton et al. [14]. A polymeric silver complex with pyridine-3,5-dicarboxylate and pyrimidine showed good antimicrobial activity with minimum inhibitory concentration (MIC) values of 64–256 µg/mL [15]. Three new silver coordination polymers were prepared from silver oxide: 1,3,5-triaza-7-phosphaadamantane and the dicarboxylic acids 3-phenylglutaric, phenylmalonic, or 3,3-dimethylglutaric acid [16]. The polymers were tested against *E. coli*, *P. aeruginosa*, and *S. aureus*, showing MICs of 11–23 nmol/mL. The biological activities of copper, zinc, and nickel coordination compounds with dicarboxylic acids 2,2′-thiodiacetic (H_2_tda), 3,3′-thiodipropionic, 3,3′-dithiodipropionic, and fumaric acid are listed in a review [17]. The inhibition zones for *B. subtilis*, *S. aureus*, *E. faecalis*, *E. coli*, *K. pneumoniae*, and *P. aeruginosa* on the complexes [(phen)_2_Cu(*μ*-tda)Cu(phen)](ClO_4_)_2_ and [Cu(phen)(tda)]·2H_2_O, (phen = 1,10-phenanthroline) were determined [18]. The interaction with albumins (bovine serum albumin—BSA—and human serum albumin—HSA) and the CT-DNA binding of the complex {[Cu_2_(L^4^)_2_(fu)]·(H_2_O)·(MeOH)}*_n_*, where HL^4^ = *(E)*-2-((1-hydroxybutan-2-ylimino)methyl)phenol (a Schiff base formed by the condensation of salicylaldehyde with 2-amino-1-butanol), was reported by Paul et al. [19].

In our previous study, we prepared and characterized six nickel and copper mixed ligand complexes, including two complexes studied in the present paper [20]. The complexes, [Ni(tda)(phen)(H_2_O)]·3H_2_O (complex No. 1), [Ni(tda)(1,3-pn)(H_2_O)]·H_2_O (complex No. 2), [Ni_2_(μ-tda)_2_(1,2-pn)_2_] (complex No. 3), [Cu_2_(μ-tdp)(pmdien)_2_(H_2_O)_2_](ClO_4_)_2_·H_2_O (complex No. 4), [Cu_2_(μ-fu)(pmdien)_2_(H_2_O)_2_](ClO_4_)_2_ (complex No. 5), [Cu_2_(*μ*-dtdp)(pmdien)_2_(H_2_O)_2_](ClO_4_)_2_ (complex No. 6), and (H_2_tda = 2,2′-thiodiacetic acid, H_2_tdp = 3,3′-thiodipropionic acid, 1,2-pn = 1,2-diaminopropane, 1,3-pn = 1,3-diaminopropane and phen = 1,10-phenanthroline), were evaluated as antibacterial agents. This work was focused on nickel complexes containing a tda dianion bridge and N-N donor ligands. It was found that these bidentate N-N ligands have an impact on the nuclearity of nickel complexes. The sulfur atom of tda is involved in coordination with the central atom. The copper complexes are dinuclear with different dicarboxylate bridges and the tridentate N donor ligand pmdien. Although there is a sulfur atom in tdp or dtdp dianions, they do not interact with central copper atoms. The solution stability of complexes was studied using UV-Vis spectroscopy. The antibacterial properties of these complexes were evaluated against Gram-positive (*E. faecalis*, *S. aureus*) and Gram-negative (*P. aeruginosa*, *E. coli*) bacteria. The MIC values were found: the most potent were copper complex No. 5 (MIC = 4.22, 1.05, 2.11, 8.44 mg/L, respectively) and complex No. 6 (MIC = 8.44, 8.44, 1.05, 16.88 mg/L, respectively) [20]. As for each promising compound, the potential inhibitory effect of these antimicrobial drugs on the activity of the most important human liver drug-metabolizing cytochromes P450 (CYP) was determined, especially CYP1A2, CYP2A6, CYP2B6, CYP2C8, CYP2C9, CYP2C19, CYP2D6, CYP2E1, and CYP3A4.

CYPs, the key enzymes of the first phase of drug metabolism [21], are localized in many tissues of the human organism (e.g., liver, lungs, brain, etc.). CYP enzymes are typically localized in a cell in the membrane of the endoplasmic reticulum [22]. CYPs’ main function is the formation of more polar metabolites of drugs, either by inserting the polar group into the parent molecule (e.g., hydroxylation) or by liberating a present functional group (e.g., demethylation of a methoxy group). In pharmacology, focus is given to CYPs that are present in the human liver, the most important organ in the biotransformation of drugs [21]. The possible effect of a promising drug on the activity of these enzymes may be a reason for unwanted drug–drug interactions with concomitantly taken medications.

## 2. Results

### 2.1. Complex Preparation

[Cu_2_(μ-fu)(pmdien)_2_(H_2_O)_2_](ClO_4_)_2_ (complex No. 5) and [Cu_2_(*μ*-dtdp)(pmdien)_2_(H_2_O)_2_](ClO_4_)_2_ (complex No. 6) were prepared according to a published procedure [20]. The successful preparation process in this study was proved using elemental analysis and infrared spectroscopy. Complex No. 5 was recrystallized from water, as well as from methanol and dark blue crystals of the same composition, showing very good stability of the complex in the mentioned solvents. The structure of complex No. 6 was presented by us in [20]. The structure of complex No. 5 was measured again (the structure was previously solved with worse parameters by Mautner [23]), and our new single crystal data were deposited under CCDC 2378750. Tables with crystal data, structure refinement, and selected bond lengths and angles are presented in ESI (Appendix A). The molecular structures are depicted in Figure 1. It can be seen that the binuclear copper complexes differ in the dicarboxylato bridge. The coordination number on copper is five. All the copper centers are coordinated by three nitrogen atoms of tridentate chelating ligand pmdien and two oxygen atoms. One oxygen is from a water molecule, and the second one is from a monodentately bonding carboxylato group. The fumarato bridge contains a double bond, whereas the dithiodipropionato bridge is longer, with two sulfur atoms. Neither the double bond nor the sulfur atoms are involved in coordination with central atoms. Two perchlorate anions in each molecule are not involved in coordination. There are only hydrogen bonds among the oxygen atoms of the perchlorate group and hydrogens of the coordination cation.

Moreover, the composition of the compounds prepared in bulk was verified through powder X-ray diffraction (see Appendix A). The diffraction peaks calculated from single crystal diffraction data are in good agreement with those found in the powder diffraction patterns.

### 2.2. Mass Spectrometry of the Tested Complexes

Mass spectrometry was used to further characterize the behavior of the complexes in a water solution. The mass spectra (Appendix A) display intense peaks at *m*/*z* = 686.96 and 780.91 for 5 and 6, respectively. These peaks correspond to the binuclear molecular ions of the composition of [Cu_2_(fu)(pmdien)_2_(ClO_4_)H^+^]^+^ and [Cu_2_(dtdp)(pmdien)_2_(ClO_4_)H^+^]^+^ for complexes No. 5 and 6, respectively. The simulated paterns agree well with the measured data (Appendix A).

Thus, we can deduce that in a solution, the complexes remain almost unchanged, except for the coordinated water molecules, which are either exchanged for a perchlorate ion or a formed perchlorate adduct. The adduct formation with the perchlorate ions is well known [24,25]. There are also peaks in the spectrum of 5 at *m*/*z* = 630.80, 517.18, and 236.08, corresponding to fragments of the complex. Similarly, in the spectrum of 6, there are peaks at *m*/*z* = 445.04, 335.00 and 236.09.

### 2.3. Spectral Study of the Interactions of the Tested Complexes with HLM CYP Enzymes

Spectral studies in the Soret region were used to detect the substrate-induced difference spectra of human liver microsomal (HLM) CYPs [26]. From the inspection of the course of the difference spectra for the tested compound, it follows that complexes No. 5 and No. 6 bind to HLM CYP enzymes as substrates. The spectral change corresponds to the formation of a maximum formed at about 380 nm and a minimum at about 417 nm (Figure 2). The values of the corresponding spectroscopic binding constant (K_S_) for the tested compounds were 39.50 ± 10.06 µmol·L^−1^ and 44.86 ± 9.88 µmol·L^−1^, respectively. These values indicate the relatively specific binding of these complexes on CYPs.

### 2.4. Inhibition Study of the Tested Complexes with HLM CYP Enzymes

In the present study, the inhibitory effects of complexes No. 5 and No. 6 on the enzyme activities of nine HLM CYP enzymes were investigated. At first, the enzyme kinetics of individual CYP forms were determined using standard substrates. Additionally, experiments assessing the inhibition of enzyme activities of nine CYP forms by complexes No. 5 and No. 6 were performed. As for all potential drugs, the possibility of interactions with the xenobiotic metabolizing CYP enzyme system should be checked. After application, any drug is distributed throughout the body. This also happens in the case of topically applied drugs, where a contribution of CYP enzymes, known to be present in the skin, to the drug metabolism takes place [27].

Figure 3 shows the summarized results of the inhibition experiments (Figure 3a for complex No. 5; Figure 3b for complex No. 6). Both complexes show prominent inhibitory effects on the enzyme activities of CYP1A2, CYP2A6, CYP2B6, CYP2C8, CYP2C9, CYP2C19, CYP2D6, CYP2E1, and CYP3A4. The tested complexes inhibit the activities of CYP2C8, CYP2D6, and CYP3A4 considerably to 0% of the corresponding control; this is at the highest concentration (50 µmol·L^−1^). The inhibition concentration (IC_50_) values for each experiment were determined as well (see Table 1).

## 3. Discussion

The interaction of drug-metabolizing enzymes, i.e., primarily CYPs, is one of the necessary steps in the evaluation of the suitability of a drug for clinical application [28,29,30,31,32]. Inhibition of a CYP-mediated metabolism of a drug through the one-way interaction with another simultaneously applied drug leads to drug accumulation and an increase in drug concentrations up to toxic values.

To obtain information on the possible drug–drug interactions at the level of the CYP-mediated metabolism, the effect of both complexes, tested on nine human liver microsomal CYP activities, was investigated. As documented by means of difference spectroscopic studies, both complexes were shown to be able to interact with microsomal CYP enzymes (Figure 2). The observed spectral changes can be interpreted as the binding of both compounds to the CYP substrate binding site, as the course of the spectral changes show an increase in absorption at about 380 nm, accompanied by a spectral minimum at about 417 nm. These spectral changes are typical of the formation of a complex of interacting compounds with the CYP enzyme (Type I binding spectra [26]). The next step was a study of the inhibition of CYP activities that are characteristic of individual forms of CYP using their prototype substrates (Table 2 and Figure 3). In line with the spectral studies, both complexes were inhibiting the corresponding enzyme actvities in a concentration-dependent manner. The strongest inhibition in a micromolar range occurred with CYP2C8 (IC_50_ for both complexes is about 3.5 µmol·L^−1^), CYP2C19 (IC_50_ is for complex No. 5 about 2.5 µmol·L^−1^ and for complex No. 6 about 6.4 µmol·L^−1^), and CYP3A4 (IC_50_ for both complexes is about 3.7 µmol·L^−1^).

Both the mentioned CYP2C forms, as well as CYP3A4, have been found to be involved, e.g., in the metabolism of the often-prescribed antiplatelet drug clopidogrel. Clopidogrel is often used in the prevention of atherothrombotic episodes in patients with coronary artery disease who have undergone stent implantation [29]. At the first step, CYP2C19 catalyzes oxidation of clopidogrel, which is de facto an activation of the clopidogrel prodrug. The second reaction, also a CYP-catalyzed reaction, is the formation of a nonactive metabolite; both CYP2C forms and CYP3A4 participate in this biotransformation [42]. Accordingly, the inhibition of CYP2C19 enzyme activity could be responsible for the deficient activation of the prodrug and its decreased efficacy; similarly, inhibition of the activity of other mentioned CYPs could lead to deficient drug metabolism and accumulation of the drug in the human organism, resulting in an increase in drug plasma levels and the manifestation of adverse or even toxic effects.

CYP3A4, in other words, plays a major role in the drug metabolism, as it participates in the biotransformation of about 50% of clinically used drugs [43], such as the hypolipidemic agent atorvastatin, antidepressant citalopram, antibiotics clarithromycin or erythromycin, anticonvulsant diazepam, opiate morphine, etc. Oral application of both tested complexes (which inhibit CYP3A4) may hence open the possibility of drug–drug interactions with concomitantly taken drugs.

In contrast, topical application of the studied complexes, as potential antimicrobial agents, lowers the possibility of adverse effects, as the plasma levels of a topically administered drug are expected to be lower than those of the same drug administered orally. An advantage of topical application is an avoidance of the first pass effect, which is predominantly caused by the metabolism of drugs by liver enzymes such as CYP3A4 [44].

In higher concentrations, the studied complexes can also influence the activities of other tested liver microsomal CYP enzymes, which may result in adverse effects when other drugs that are metabolized by the same CYP enzymes are administered in parallel. For instance, an unwanted interaction on the basis of metabolism at the CYP1A2 might happen with drugs such as the antipsychotic clozapine, analgesic lidocaine, or antihypertensive drug propranolol. CYP2C9, which is also effectively inhibited by both complexes studied here, is known for its metabolism of anticoagulant warfarin. CYP2D6 has been shown to be the key enzyme in the metabolism of antidepressants, e.g., amitriptyline or chlorpromazine. Halothane and isoflurane are commonly applied inhaled anesthetics that are metabolized via CYP2E1 [43]. With regard to the results described here, the topical application of these potential antimicrobial agents could be a relatively safe way of administration.

## 4. Materials and Methods

### 4.1. Chemicals and Reagents

Complex No. 5—[Cu_2_(μ-fu)(pmdien)_2_(H_2_O)_2_](ClO_4_)_2_ and No. 6—[Cu_2_(μ-dtdp)(pmdien)_2_(H_2_O)_2_](ClO_4_)_2_ were synthesized according to previously described procedures [20]. Briefly, copper perchlorate hexahydrate (0.37 g) was dissolved in methanol (50 mL), and pmdien (0.2 mL) was slowly added while stirring on a magnetic stirrer. The potassium salt of fumaric (0.1 g) resp and dithiodipropionic acid (0.14 g) dissolved in methanol (25 mL) was poured into the reaction mixture. After 2 days, transparent crystals of potassium perchlorate were removed, and the solution was left to crystalize. Dark blue crystals of complexes were collected after a week. The complexes were recrystallized from water.

The crystal structure of complex No. 5 was determined on an XtaLAB Synergy-I diffractometer with a HyPix3000 hybrid pixel array detector and microfocused PhotonJet-I X-ray source (Cu Kα = 1.54178 Å). The X-ray powder diffraction patterns were recorded on a MiniFlex 600 X-ray difractometer (Rigaku, Austin, TX, USA) with a Bragg–Brettano arrangement using CuKα radiation. The mass spectra were obtained on an LCQ Fleet mass spectrometer (Thermo Scientific, Waltham, MA, USA) equipped with an electrospray ion source and three-dimensional (3D) ion-trap detector in the positive mode.

7-ethoxyresorufin and ethoxy-4-(trifluoromethyl)coumarin were supplied by Fluka (Buchs, Switzerland). All other chemicals were purchased from Sigma Aldrich CZ (Prague, Czech Republic). All other common laboratory chemicals of a high-performance liquid chromatography (HPLC) or analytical grade were obtained from the same company. Pooled human liver microsomes were obtained from Biopredic (Rennes, France). The microsomes were obtained in accordance with ethical regulations of the country of origin (France). They came from seventeen donors (ten males and seven females), with a protein content of 25 mg/mL and total CYP content of 14.55 µmol·L^−1^; CYP1A2, CYP2A6, CYP2B6, CYP2C8, CYP2C9, CYP2C19, CYP2D6, CYP2E1, and CYP3A4/5 enzyme activities were verified before the experiment with minimal consumption of the sample.

### 4.2. Spectral Study of the Interaction of the Complexes with HLM CYPs

The binding difference spectra of the interactions of complex No. 5 and complex No. 6 with microsomal CYP enzymes were followed according to established procedures [26,45]. The cuvette contained HLM that was diluted to a final concentration of CYP 1 µmol·L^−1^ in 50 mmol·L^−1^ potassium phosphate buffer (pH 7.4). The tested compounds were dissolved in the same buffer, and the concentration ranges in experiments were 0–183 µmol·L^−1^, obtained by diluting a 5 mmol·L^−1^ stock solution. Spectra were recorded at room temperature by repetitive scanning between 300 and 700 nm at a scan speed of 200 nm/s. The baseline of the reference cuvette contained HLM diluted to the final concentration of CYP 1 µmol·L^−1^ in 50 mmol·L^−1^ potassium buffer (pH 7.4). Spectra were recorded on a Varian 4000 UV/VIS spectrophotometer (Varian, Palo Alto, CA, USA). The absorbance change at about 417 nm was plotted against the concentration of the tested compounds. Data were analyzed using the Sigma Plot v. 8.0 graphing software (Jandel Scientific, Chicago, IL, USA).

### 4.3. Enzyme Assays

All microsomal CYP activities were assayed according to well-established protocols.

The following enzyme assays were performed to determine the activities of specific CYP enzymes: CYP1A2, 7-ethoxyresorufin *O*-deethylation [33]; CYP2A6, coumarin 7-hydroxylation [34]; CYP2B6, 7-ethoxy-4-(trifluoromethyl) coumarin (EFC) *O*-deethylation [35]; CYP2C8 paclitaxel 6α-hydroxylation [36]; CYP2C9, diclofenac 4′-hydroxylation [37]; CYP2C19, diazepam *N*-desmethylation [38]; CYP2D6, bufuralol 1′-hydroxylation [39]; CYP2E1, chlorzoxazone 6-hydroxylation [40]; and CYP3A4/5, testosterone 6β-hydroxylation [41].

The incubation mixtures contained 100 mmol·L^−1^ potassium phosphate buffer (pH 7.4), an NADPH-generating system (0.8 mmol·L^−1^ NADP^+^, 5.8 mmol·L^−1^ isocitrate, 0.3 unit/mL of isocitrate dehydrogenase, and 8 mmol·L^−1^ MgCl_2_), HLM, and an individual probe substrate. The assay conditions are listed in Table 2.

For the determination of metabolites formed from specific substrates, a Prominence LC-20A HPLC system (Shimadzu, Kyoto, Japan) with UV detection or fluorescence detection was used (see Table 2). As a rule, reversed-phase chromatography was applied with Merck (Darmstadt, Germany) LiChroCART 250 × 4 mm (i.d.) cartridges packed with a LiChrospher RP-18 silica gel (5 µm particle size) with C-18 reversed-phase properties; a 4 × 4 mm (i.d.) guard column was also used. For the composition of the mobile phase in the respective analyses, see the original reference.

### 4.4. Enzyme Inhibition Studies

Initially, for each enzyme assay, a preliminary experiment was carried out to determine K_m_ and V_max_ to obtain the appropriate concentration of the specific substrates for the inhibition experiments (the substrate concentration was chosen in the range corresponding to the value of the K_m_ for each particular reaction). Data were analyzed using the Sigma plot software, as mentioned above, with spectral studies. Inhibition experiments were performed with a concentration range of the tested complexes from 0.78 to 50 µmol·L^−1^. The concentration range used here is in accordance with the MIC (see above) and is adequate. The experimental conditions were the same as for the determination of individual CYP activities (Table 2); the assays were performed in duplicates, with individual values differing by less than 20%. Preincubation of the reaction mixtures with the tested complexes for 30 min at 37 °C was carried out in all assays. In all cases, the inhibition of individual CYP activities was evaluated by plotting the respective remaining activity against the inhibitor concentration.

## 5. Conclusions

In summary, [Cu_2_(μ-fu)(pmdien)_2_(H_2_O)_2_](ClO_4_)_2_ (complex No. 5) and [Cu_2_(μ-dtdp)(pmdien)_2_(H_2_O)_2_](ClO_4_)_2_ (complex No. 6) were studied because of their possible use as antimicrobial agents. The effect of both complexes, tested on nine human liver microsomal CYP activities, was investigated. All enzyme activities were inhibited in a concentration-dependent manner. The results show strongly inhibited enzyme activity of CYP2C8, CYP2C19, and CYP3A4. In these cases, the presence of adverse effects due to drug–drug interactions with concomitantly used drugs cannot be excluded. This is why the topical application of these potential antimicrobial agents could be a relatively safe way of administration, as the presumed plasma levels could be relatively low.

## Figures and Tables

**Figure 1 pharmaceuticals-17-01194-f001:**
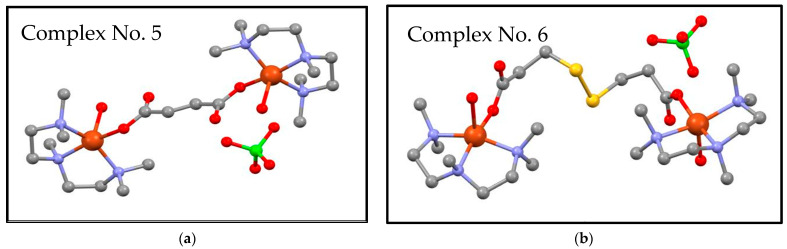
Molecular structures of tested complexes No. 5 (**a**) ([Cu_2_(μ-fu)(pmdien)_2_(H_2_O)_2_](ClO_4_)_2_) and No. 6 (**b**) ([Cu_2_(μ-dtdp)(pmdien)_2_(H_2_O)_2_](ClO_4_)_2_) [20]. Hydrogen atoms are not shown, and only one perchlorate anion is depicted. Orange stands for copper, blue for nitrogen, red for oxygen, yellow for sulfur, green for chlorine, and gray for carbon.

**Figure 2 pharmaceuticals-17-01194-f002:**
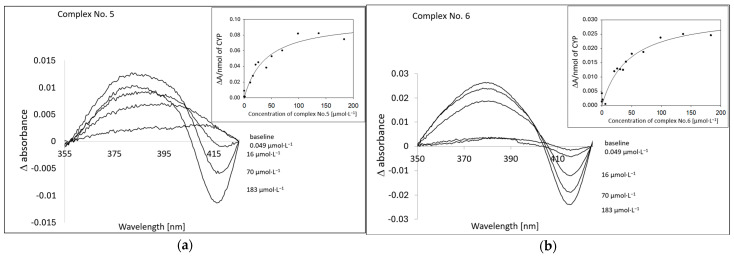
Spectral determination of the interaction between tested complexes No. 5 (**a**) and No. 6 (**b**) and CYP in HLM. The insert shows the plot of the spectral difference (ΔA/nmol of CYP) at about 417 nm vs. the complex concentrations for the determination of K_S_ (see text). The concentration of CYP in microsomal preparation, 1 µmol·L^−1^ in 50 mmol·L^−1^ potassium phosphate buffer (pH 7.4).

**Figure 3 pharmaceuticals-17-01194-f003:**
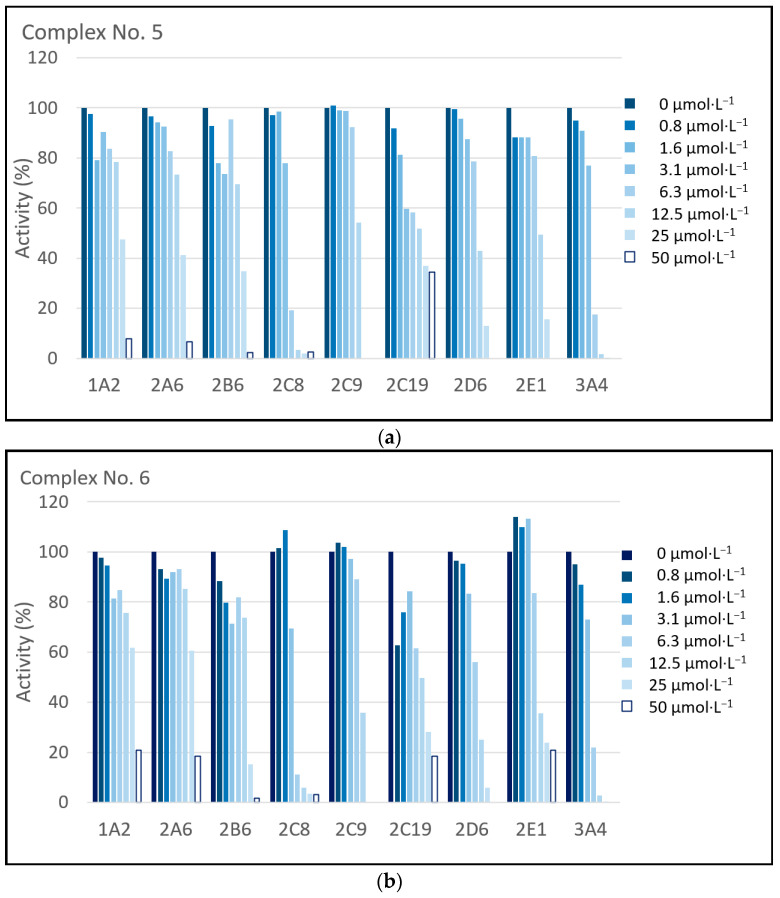
The effects of complexes No. 5 (**a**) and No. 6 (**b**) on the catalytic activities of CYP1A2, CYP2A6, CYP2B6, CYP2C8, CYP2C9, CYP2C19, CYP2D6, CYP2E1, and CYP3A4 enzymes in HLM. Inhibition of catalytic activities is expressed as the activity remaining relative to the control (100%, without complexes) in percent. The concentration of complexes in the reaction mixture ranged from 0 to 50 µmol·L^−1^. For experimental details, see Methods (Section 4.3). The assays were performed in duplicates, with individual values differing less than 20%.

**Table 1 pharmaceuticals-17-01194-t001:** Inhibition concentrations (IC_50_) of complex No. 5 and No. 6 as determined in our experiments.

	CYP1A2	CYP2A6	CYP2B6	CYP2C8	CYP2C9	CYP2C19	CYP2D6	CYP2E1	CYP3A4
	IC_50_ [µmol·L^−1^]								
Complex No. 5	24.97 ± 0.22	24.85 ± 0.25	24.81 ± 0.33	3.66 ± 0.19	12.59 ± 0.06	2.53 ± 0.59	12.37 ± 0.21	12.52 ± 0.18	3.72 ± 0.23
Complex No. 6	25.17 ± 0.29	25.13 ± 0.13	13.29 ± 0.54	3.40 ± 0.10	12.26 ± 0.11	6.43 ± 0.63	6.33 ± 0.22	6.59 ± 0.18	3.62 ± 0.27

Mean ± S.D.

**Table 2 pharmaceuticals-17-01194-t002:** Incubation conditions for individual CYP assays used in inhibition study.

CYP	Substrate Concentration (µmol·L^−1^)	Substrate	Reaction Catalyzed by CYP	Content CYP (nmol)	Reaction Volume (µL)	Quench Reagent	Method of Detection	References
1A2	1.3	7-ethoxyresorufin	*O*-deethylation	35	100	200 µL methanol	fluorescence	[33]
							ex. 535 nm; em. 585 nm	
2A6	15	coumarin	7-hydroxylation	35	100	200 µL methanol	fluorescence	[34]
							ex. 325 nm; em. 450 nm	
2B6	15	EFC	*O*-deethylation	35	100	200 µL methanol	fluorescence	[35]
							ex. 410 nm; em. 510 nm	
2C8	50	paclitaxel	6α-hydroxylation	70	200	50 µL ice cold	UV, 230 nm	[36]
						acetonitrile		
2C9	16	diclofenac	4′-hydroxylation	35	200	50 µL 96% ACN/	UV, 280 nm	[37]
						4% CH_3_COOH		
2C19	150	diazepam	*N*-desmethylation	70	200	100 µL acetonitrile	UV, 236 nm	[38]
2D6	15	bufuralol	1′-hydroxylation	67	200	20 µL 70% HClO_4_	fluorescence	[39]
							ex. 252 nm; em. 302 nm	
2E1	30	chlorzoxazone	6-hydroxylation	160	1000	50 µL 42.5% H_3_PO_4_	UV, 287 nm	[40]
3A4	100	testosterone	6β-hydroxylation	100	500	100 µL 1M Na_2_CO_3_/	UV, 245 nm	[41]
						2M NaCl		

## Data Availability

The CIF file containing the full crystallographic data (CCDC 2378750) can be obtained free of charge via http://www.ccdc.cam.ac.uk/conts/retrieving.html (or from the CCDC, 12 Union Road, Cambridge CB2 1EZ, UK; Fax: +44-1223-336033; E-mail: deposit@ccdc.cam.ac.uk). Research data are available upon request to the authors (A.Š. and P.K.).

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
