# Peer review of "In Vitro* Interaction of Binuclear Copper Complexes with Liver Drug-Metabolizing Cytochromes P450"

_pharmaceuticals, 2024, doi:10.3390/ph17091194_

Round 1

Reviewer 1 Report

Comments and Suggestions for Authors

MS number: pharmaceuticals-3173979

Type of manuscript: Article
Title: In vitro interaction of binuclear copper complexes with liver drug-metabolizing cytochromes P450
Authors: Alena Špičáková *, Zuzana Horáčková, Pavel Kopel *, Pavel Anzenbacher

This paper describes the two known copper(II) binuclear complexes with different carboxylate bridging and their biological activity. This issue is still interesting in the current inorganic chemistry field. However, I found some questionable points; therefore, I strongly requested major revisions to the next step.

Abstract and others

The coordinated nitrogen should be italics such as “N-…”

Introduction

You did not cite the important review article (Chem. Rev. 2014, 114, 3659−3853) by Prof. Solomon for “It is well known that copper binds to enzymes and is necessary for redox processes, blood, and hormone changes.”.

You did not mention when copper levels are high and when they are low. Please note the typical diseases for both types such as Wilson and Menkes diseases.

Results

The way the copper complexes are described is very vague and confusing. Complexes 5 and 6 were synthesized in reference 17, and the structure of complex 5 seems to be in reference 17 and complex 6 in reference 20. But this is not what I could read from the main text. However, this comes from the result of my research by checking the references by myself. Please describe it so that I can understand.

X-ray structure

I think you should permit the original references (17 and 20) for the descriptions of the X-ray structures in Figure 1 (your text).

Powered X-ray structure

Why does Figure 1 describe two types?

The description of the supporting material should be Figure S1, etc.

Mass spectroscopy

You should add the appropriate simulation patterns in the figure.

Others

Should there be no description of SI in the text?

Comments on the Quality of English Language

I think some improves need for the publication. 

Author Response

MS number: pharmaceuticals-3173979

Type of manuscript: Article
Title: In vitro interaction of binuclear copper complexes with liver drug-metabolizing cytochromes P450
Authors: Alena Špičáková *, Zuzana Horáčková, Pavel Kopel *, Pavel Anzenbacher

This paper describes the two known copper(II) binuclear complexes with different carboxylate bridging and their biological activity. This issue is still interesting in the current inorganic chemistry field. However, I found some questionable points; therefore, I strongly requested major revisions to the next step.

Answer: 

Dear Reviewer,

Thank you very much for your comments. We have made our best to prepare amended of our manuscript according to the comments.

Abstract and others

The coordinated nitrogen should be italics such as “N-…”

Answer: Small typos in the text have been corrected, e.g. N- vs N- etc.

 Introduction

You did not cite the important review article (Chem. Rev. 2014, 114, 3659−3853) by Prof. Solomon for “It is well known that copper binds to enzymes and is necessary for redox processes, blood, and hormone changes.”.

Answer: Thank you very much for your comment regarding the excellent and comprehensive review by Prof. Solomon et al. We have listed it in the references.

 You did not mention when copper levels are high and when they are low. Please note the typical diseases for both types such as Wilson and Menkes diseases.

 Results

The way the copper complexes are described is very vague and confusing. Complexes 5 and 6 were synthesized in reference 17, and the structure of complex 5 seems to be in reference 17 and complex 6 in reference 20. But this is not what I could read from the main text. However, this comes from the result of my research by checking the references by myself. Please describe it so that I can understand.

Answer: We have rephrased and reworded the first paragraph in the Introduction and we also mentioned the typical diseases such as Wilson´s and Menkes´diseases. We have rephrased the paragraph in the Results to be more clear as well. Thank you for your insight.

X-ray structure

I think you should permit the original references (17 and 20) for the descriptions of the X-ray structures in Figure 1 (your text).

 Powered X-ray structure

Why does Figure 1 describe two types?

 The description of the supporting material should be Figure S1, etc.

Answer: Regarding your comment about X-ray structure, the structures have been described in the text and one figure there was accidentally added, it has been removed. Numbering of Figure S1 has been changed, thank you for your notice.

 Mass spectroscopy

You should add the appropriate simulation patterns in the figure.

 Others

Should there be no description of SI in the text?

Answer: On the basis of your comment, the figure of the mass spectroscopy is added in ESI and SI description has been added.

Reviewer 2 Report

Comments and Suggestions for Authors

The authors study the in vitro interaction of binuclear copper complexes with P450 enzymes. The effect of two tested copper complexes on nine human liver microsomal CYP activities was investigated. The CYP-related experimental work is well performed; however, there is some confusion in the description between previously obtained data and the original data. Authors asked to include more concrete data from the obtained results in the abstract.

Critical issues:

  In the Abstract, lines 20-25, two sentences are too complicated to understand. Please rephrase them. Be careful to clearly outline the original data described in this paper and avoid mixing it with previously obtained data (e.g., solubility, chemotherapeutic potential, antibacterial activity).

  Abstract: One clear, conclusive sentence is needed.

  Introduction, line 63: Define the abbreviation "MIC."

Results, lines 110-122: This section contains a mix of previously obtained data and original data. Please move the previously obtained data to the Introduction or Discussion, and focus on reporting new/original data in the Results section.

Authors mention Fig. S1 in line 119, do they mean Figure 1 AS in page 10? The same for Fig. S2. Why are the "AS" and "BS" labels used for Figures 1 and 2? Are these figures intended to be supplemental?

  Results, lines 137-139: Data in the text must be accompanied by references to the corresponding figures.

  In Figures 2(a) and 2(b), the characters in the inset graphs are too small and pixelated, making them almost unreadable.

  In Figure 2(b), within the absorbance difference spectrum, the 70 and 163 μM concentrations show a significant difference in the spectrum shift. However, in the inset graph, all concentrations over 40 μM correspond to the same value of the binding constant (the horizontal right part of the curve). Please double-check the data for this incongruence or provide a comment in the text on the actually presented observed data. The similar spectra in Figure 1(a) seem completely correct.

  Figure 2 legend, line 151: “...at about 417 nm vs. complex concentrations…” was used for Ks plots. In Methods, line 272: “Absorbance change at about 380 nm was plotted against the concentration of the compounds tested.” Please double-check which wavelength was actually used for Ks.

  Methods, line 290: "Table 1" is mentioned, but should this be "Table 2"?

  Methods, line 292: "5 mm particle size" is mentioned. Should this be "5 μm"?

  Discussion: Please explain (or speculate on) the following: On one hand, the Ks concentration for complex 1 is 34 μM (line 147), and in Figure 2(a), complex 1 continues to bind the enzyme up to 100 μM. On the other hand, in terms of inhibition, 50 μM of complex 1 almost completely inhibits individual substrate bindings in Figure 3.

Author Response

The authors study the in vitro interaction of binuclear copper complexes with P450 enzymes. The effect of two tested copper complexes on nine human liver microsomal CYP activities was investigated. The CYP-related experimental work is well performed; however, there is some confusion in the description between previously obtained data and the original data. Authors asked to include more concrete data from the obtained results in the abstract.

Answer: 

Dear Reviewer,

Thank you sincerely for your comments, which helped us to improve the paper. We have made adjustments of the manuscript according to the these comments.

Critical issues:

  • In the Abstract, lines 20-25, two sentences are too complicated to understand. Please rephrase them. Be careful to clearly outline the original data described in this paper and avoid mixing it with previously obtained data (e.g., solubility, chemotherapeutic potential, antibacterial activity).
  • Abstract: One clear, conclusive sentence is needed.

Answer: We have rephrased and reworded the paragraph in the Abstract, we apologize for formulations and these sentences have been reformulated and rephrased to make the Abstract more clear. Thank you for your insights.

  • Introduction, line 63: Define the abbreviation "MIC."

Answer: Definition of abbreviation has been defined at the first place where i tis mentioned in the text. Thank you.

  • Results, lines 110-122: This section contains a mix of previously obtained data and original data. Please move the previously obtained data to the Introduction or Discussion, and focus on reporting new/original data in the Results section.

Answer: We have rephrased the paragraph in the Results to be more clear as well and number of the figure has been corrected. Thank you for your insight.

Authors mention Fig. S1 in line 119, do they mean Figure 1 AS in page 10? The same for Fig. S2. Why are the "AS" and "BS" labels used for Figures 1 and 2? Are these figures intended to be supplemental?

  • Results, lines 137-139: Data in the text must be accompanied by references to the corresponding figures.

Answer: All figures marked S1, S2 and S3 at the end of the article have been provided as supplementary material.

  • In Figures 2(a) and 2(b), the characters in the inset graphs are too small and pixelated, making them almost unreadable.
  • In Figure 2(b), within the absorbance difference spectrum, the 70 and 163 μM concentrations show a significant difference in the spectrum shift. However, in the inset graph, all concentrations over 40 μM correspond to the same value of the binding constant (the horizontal right part of the curve). Please double-check the data for this incongruence or provide a comment in the text on the actually presented observed data. The similar spectra in Figure 1(a) seem completely correct.

Answer: Figures 2 have been redesigned to be more readable and another corrections of the text have been made on the basis of your observations. Values of mentioned Ks have been recalculated and noted into the text. Thank you again for your comment.

  • Figure 2 legend, line 151: “...at about 417 nm vs. complex concentrations…” was used for Ks plots. In Methods, line 272: “Absorbance change at about 380 nm was plotted against the concentration of the compounds tested.” Please double-check which wavelength was actually used for Ks.

Answer: Actually wavelenght of the specral diffrence at 417 nm was used. Thank you.

  • Methods, line 290: "Table 1" is mentioned, but should this be "Table 2"?
  • Methods, line 292: "5 mm particle size" is mentioned. Should this be "5 μm"?

Answer: Correct number is Table 2, it was also corrected in the text and the unit of particle size was corrected as well (to micrometer). Thank you.

  • Discussion: Please explain (or speculate on) the following: On one hand, the Ks concentration for complex 1 is 34 μM (line 147), and in Figure 2(a), complex 1 continues to bind the enzyme up to 100 μM. On the other hand, in terms of inhibition, 50 μM of complex 1 almost completely inhibits individual substrate bindings in Figure 3.

Answer: 

Figures 2 and 3 describe two different effects. Figure 2 describes an effect of interaction of compounds studied on an „average“ CYP present in the microsomal homogenate, whereas Figure 3 and Table 1 describe an inhibitory effect of tested substances on enzyme activities of individual CYP forms.

Differences in the Ks and inhibition characteristics (IC50) may be explained as follows: the spectral disociation constant Ks should be related or, apparently reflected, in the characteristics of inhibition of CYP enzyme activities. In fact, without an effect seen in the absorption spectra, clearly no inhibition of enzyme activity could be present. However, as these two events (althought related) are of diffrenet principle, the Ks values and IC50 for inhibition may be (and are) generally different.

Round 2

Reviewer 1 Report

Comments and Suggestions for Authors

Okay, it will be ready to be published in this journal.

Comments on the Quality of English Language

Please check your text again and again for the publication.